# Assessment of the Physiological Response and Productive Performance of Vegetable vs. Conventional Soybean Cultivars for Edamame Production

**Laura Matos Ribera** [1,*], **Eduardo Santana Aires** [1], **Caio Scardini Neves** [1], **Gustavo do Carmo Fernandes** [1], **Filipe Pereira Giardini Bonfim** [1], **Roanita Iara Rockenbach** [2], **João Domingos Rodrigues** [3] **and Elizabeth Orika Ono** [3]

[1] Department of Horticulture, School of Agronomy, São Paulo State University, Botucatu 18618-000, Brazil; e.aires@unesp.br (E.S.A.); caio.scardini@unesp.br (C.S.N.); gustavodocfernandes@gmail.com (G.d.C.F.); filipe.giardini@unesp.br (F.P.G.B.)

[2] Department of Agronomy, Mato Grosso do Sul State University, Aquidauana 79200-000, Brazil; roanita_rockenbach@hotmail.com

[3] Department of Botany, Institute of Biosciences, São Paulo State University, Botucatu 18618-000, Brazil; joao.domingos@unesp.br (J.D.R.); elizabeth.o.ono@unesp.br (E.O.O.)

[*] Correspondence: laura.ribera@unesp.br; Tel.: +55-119-9779-3308

**Abstract:** Because there is a close relationship between plant physiological response and crop performance, the current study aims to evaluate the photosynthetic efficiency and productive performance of vegetable versus conventional soybean cultivars for edamame production. The study was conducted at the School of Agriculture (FCA UNESP), Botucatu-SP, Brazil. The treatments in this study included soybean cultivars: vegetable-type BRS 267, vegetable-type BRSMG 790A), and type soybean cultivar grain 58HO124 EP RR, with ten repetitions per treatment in a completely randomized block design. Gas exchange and the response of the cultivars to light were evaluated for photosynthetic characterization. The first pod insertion height, plant height, number of pods per plant, and production in immature grains were all assessed for cultivar productive performance. The type of soybean cultivar grain and vegetable types of soybean showed different behaviors on physiology and yield. The vegetable-types BRS 267 and BRSMG 790A had the highest average for first pod insertion height. The vegetable type BRS 267, whose photoassimilates were designated for vegetative development, had the greatest average plant height. The conventional type 58HO124 EP RR showed greater assimilation of $CO_2$; however, the photoassimilates were directed to floral emission because such features are inherent in its ability. Finally, vegetable-type BRSMG 790A produced the most immature grains per plant while also having the greatest first pod insertion height, being the best in converting photoassimilates for edamame production.

**Keywords:** edamame; photosynthesis; production potential; soybean type

## 1. Introduction

Vegetable soybean or edamame, which is popular in Asian nations [1], may be eaten in a variety of ways, including stews, chips [2], snacks, soups, and salads [3]. The pods are big in the consumed part [4], and the grains are bright green with a yellow or grey hilum [5], with organoleptic properties that stimulate ingestion, such as sweet seeds [6] and less beany taste [7]; these vary throughout the development of the plant [8].

In terms of how it is grown, sowing time, fertilizer recommendations, and the need to manage pests and diseases, vegetable soybeans are comparable to conventional soybeans [9]. Nevertheless, the harvest period varies by crop; for vegetable soybeans, it occurs when the pods are 80% mature and the humidity is approximately 65% [10]. Furthermore, the appearance, food quality, and nutritional content of the vegetable soybean are used to

evaluate overall quality [11], characteristics that are different from those evaluated in type soybean cultivar grain soybeans.

In terms of nutritional properties, the soybean crop has high water [12] and protein contents and is healthy and accessible to the population [13]; it also has low oil content [14] and features such as the lack of lipoxygenase enzyme, low Kunitz trypsin content [15], and substantial isoflavone content [16]. This is aside from providing the market with desirable qualities such as pleasing appearance, flavor, and texture, which are all connected to palate perceptions [5].

As essential as the cultivars' genetic characteristics, crop management must be done in a natural way, supporting the development of healthy and rustic plants with a high production rate without using excessive agrochemicals [17]. Low-impact agricultural practices are an example of natural management that is used by most vegetable growers [18] to create high-value, well-accepted goods from alternative resources. Producing the crop in a low-impact organic production agricultural practice and as a horticulture plant benefits the producer by increasing production and grain size [19], as well as supplying consumers with goods that meet their needs.

Whether for cultural or traditional reasons, vegetable soybeans are being slowly introduced to the menu of Brazilians. The misinformation about the potential of soybean vegetables, in association with a lack of knowledge, constitutes one of the greatest difficulties of popularizing this crop, justifying the lack of information on commercial production and consumption of edamame, an unusual scenario since Brazil is the largest producer and exporter of type soybean cultivar grain soybeans in the world [20]. The growing demand for organic food of plant origin by the population can boost the consumption of edamame since the introduction of certain eating habits, such as vegetarianism, to the Brazilian population [21], following the global trend toward healthier habits.

Brazil is a potential producer of vegetable soybeans, having adequate soil, climate, and relief for the crop's development. Furthermore, studies that characterize the crop's physiological properties are critical as they will assist in cultivar placement in relation to the production environment, as well as provide information on genotype potential for each area.

Even though gas exchange measurements can reveal a lot about a plant's physiological state, photosynthesis is one of the most fundamental metabolic processes linked to crop biomass. Based on the presented arguments, our research aims to compare the physiological response and productive performance of vegetable vs. conventional soybean cultivars' management for edamame production.

## 2. Materials and Methods

### 2.1. Growing Conditions

The research was carried out at the Experimental Farm Lageado, which is part of the Department of Horticulture at the Faculdade de Ciencias Agronômicas Universidade Estadual Paulista (FCA/UNESP), Botucatu campus, in the state of São Paulo (22°51′ S; 48°26′ W, 786 m a.s.l.). According to the Köppen–Geiger classification, the climate is categorized as *Cfa*, i.e., warm temperate (mesothermal) humid, with average annual temperatures and precipitation of 19.5° and 1314 mm, respectively [22], and the soil is classified as Dark Red Latosol. The climate categorization is shown in Table 1.

Soil samples were taken in the 0 to 20 cm depth layer and forwarded to the Department of Soil Science (FCA UNESP) for analysis. The chemical properties revealed by the findings were as follows: pH = 4.4 in $CaCl_2$; O.M. = 21 g $dm^{-3}$; P resin = 5 g $dm^{-3}$; $Al^{3+}$ = 3 mmol $dm^{-3}$; H + Al = 36 mmol $dm^{-3}$; K = 1.4 mmol $dm^{-3}$; Ca = 10 mmol $dm^{-3}$; Mg = 3 mmol $dm^{-3}$; sum of bases (SB) = 14 mmol $dm^{-3}$; cation exchange capacity (CEC) = 51 mmol $dm^{-3}$; base saturation (BS) = 29%; S = 10 mmol $dm^{-3}$; B = 0.21 mg $dm^{-3}$; Cu = 2.3 mg $dm^{-3}$; Fe = 74 mg $dm^{-3}$; Mn = 6.3 mg $dm^{-3}$; Zn = 0.8 mg $dm^{-3}$.

**Table 1.** Climate conditions from transplanting to harvest time of vegetable and type soybean cultivar grain soybeans for edamame production. Botucatu, SP, Brazil, 2021.

| Crop Cycle | Maximum Average Temperature (°C) | Minimum Average Temperature (°C) | Accumulated Precipitation (mm) |
|---|---|---|---|
| (11/23/2020 to 12/23/2020) | 30.10 | 19.22 | 155.20 |
| (12/24/2020 to 01/24/2021) | 28.85 | 19.51 | 320.54 |
| (01/25/2021 to 02/25/2021) | 29.37 | 19.11 | 74.68 |
| (02/26/2021 to 03/08/2021) | 26.94 | 19.12 | 75.19 |

A plow and a rotavator bed former were used to prepare the area for transplanting. The soybean crop was corrected and fertilized as per the recommendation for the State of Sao Paulo [23] through the use of organic fertilizers. To achieve this, 2521.34 kg ha$^{-1}$ of 90 relative total neutralizing power (RTNP) limestone was used at planting. After a month, 1184.48 kg ha$^{-1}$ P$_2$O$_5$ was applied, followed by 13.25 kg ha$^{-1}$ of K$_2$O after a week, using rock phosphate and potassium sulfate as sources, respectively.

A total of 10 plants per meter was utilized at a population density of 200,000 plants per hectare, with a spacing between plants of 0.10 m and between lines of 0.5 m. Each plot contained 4 m$^2$, with a total of 100 soybean plants per plot, totaling 54 plants in the usable area. The seedlings were grown in polystyrene trays with 128 cells and Carolina® commercial substrate, with one seed per cell, which were previously inoculated with 2 mL of liquid inoculant (Masterfix L Premier: Stoller, Brazil), *Bradyrhizobium elkanii*, and *Bradyrhizobium japonicum* for every 0.5 kg of seeds.

Cultural treatments for low impact agricultural practices (such as manual weeding, disease, and pest control) were implemented in line with Law No. 10,831 [24] and the Technical Regulation of Normative Instruction 46 [25], which were supplemented by IN 17 [26]; therefore, the plants of cultivars BRS 267 and BRSMG 790A were cultivated according to the described laws, except for the use of the cultivar 58HO124 EP RR, a genetically modified organism (GMO). To *Microsphaera difusa* and *Peronospora manshurica*, milk syrup and Bordeaux mixture were regularly administered.

Regular application of tobacco syrup associated with manual collection was used to control recurrent insects such as the *Diabrotica spciosa*, *Euschistus heros*, *Nezara viridula*, *Edessa meditabunda*, *Piezodorus guildinii*, *Aracanthus mourei*, *Magacelis* sp., *Anticarsia gemmatalis*, and *Spodoptera frugiperda Elasmopalpus lignosellus*.

The irrigation method utilized was drip irrigation, which was first installed in the location with two lines of dripping tape in each bed, with 0.5 and 0.3 m spacing between the emitters and daily watering in the morning.

### 2.2. Experimental Design

The experimental design was used for randomized complete blocks and, therefore, the growing conditions for three soybean cultivars: BRS 267, BRSMG 790A, and 58HO124 EP RR, with ten repetitions per treatment.

The BRS 267 national cultivar is a proper vegetable-soybean cultivar. The cultivar has a determinate growth habit, yellow hilum color, medium maturity, and a hundred grain weight of 25 g. The BRSMG 790A cultivar is a conventional cultivar with potential for vegetable-soybean. The cultivar has a determinate growth habit, yellow hilum color, long maturity and a hundred grain weight of 19.6 g. Finally, cultivar 58HO124 EP RR is a type soybean cultivar grain. The cultivar has an indeterminate growth habit, light brown hilum color, medium maturity, and a hundred-grain weight of 17 g, being widely cultivated in Brazil. All cultivars are recommended for the climatic conditions of the study site.

### 2.3. Gas Exchange

The Infrared Gas Analyser portable device was used to evaluate the gas exchange of the plants (IRGA LI-6400, Li-Cor Inc., Lincoln, NE, USA). During the day, analyses were carried out on 6 plants of each cultivar that had previously been selected and standardized,

taking fully expanded leaves from the center third of the plant at 73 days after transplanting, between stages R1 and R2.

Readings were taken every 2 h from 8:00 a.m. to 6:00 p.m. It was measured with plants at the same stage of development: $CO_2$ assimilation rate ($A$, µmol $CO_2$ m$^{-2}$ s$^{-1}$), stomatal conductance ($Gs$, mol $H_2O$ m$^{-2}$ s$^{-1}$), internal carbon concentration ($Ci$, µmol $CO_2$ mol air$^{-1}$), transpiration ($E$, mol $H_2O$ m$^{-2}$ s$^{-1}$), water use efficiency (WUE, µmol $CO_2$ m$^{-2}$ s$^{-1}$/mmol$^{-1}$ $H_2O$ m$^{-2}$ s$^{-1}$) determined by the relationship between the rates of $CO_2$ assimilation, transpiration, and carboxylation efficiency ($A/Ci$ µmol $CO_2$ m$^{-2}$ s$^{-1}$/µmol m$^{-2}$ s$^{-1}$), which was obtained through the relationship between the $CO_2$ assimilation rate and internal carbon concentration.

Photosynthetic photon flux density (PPFD, µmol m$^{-2}$ s$^{-1}$) was used to confirm that the experimental conditions were consistent, and, therefore, it was standardized at each assessment period using an IRGA-coupled light-emitting diode, in compliance with the PFFD (Table 2). During the assessment, the ambient air concentration was the reference for $CO_2$ concentration, with values ranging from 380–400 µmol mol$^{-1}$ of air.

**Table 2.** Air temperature (°C), relative humidity of the air (%), and photosynthetic photon flux density (PPFD, µmol m$^{-2}$ s$^{-1}$), from 8:00 a.m. to 6:00 p.m. under field conditions. Botucatu, SP, Brazil, 2021.

| Day Time | Air Temperature °C | Relative Humidity % | PFFD µmol m$^{-2}$ s$^{-1}$ |
|---|---|---|---|
| 8:00 a.m. | 24.4 | 56.3 | 600 |
| 10:00 a.m. | 27.5 | 58.3 | 1540 |
| 12:00 a.m. | 29.2 | 52.8 | 2150 |
| 2:00 p.m. | 31.4 | 47.5 | 1570 |
| 4:00 p.m. | 30.6 | 41.8 | 1350 |
| 6:00 p.m. | 26.1 | 49.7 | 220 |

### 2.4. $CO_2$ Assimilation Response Curve as a Function of PPFD

The rate of the $CO_2$ assimilation curve ($A$, µmol $CO_2$ m$^{-2}$ s$^{-1}$) as a function of photosynthetic photon flux density (PPFD, µmol m$^{-2}$ s$^{-1}$) was obtained through the decrease of 2000 till 0 PPFD µmol m$^{-2}$ s$^{-1}$ at intervals of 200 µmol m$^{-2}$ s$^{-1}$ till 200 µmol m$^{-2}$ s$^{-1}$, and then, at intervals of 50 µmol m$^{-2}$ s$^{-1}$ for more accurate curve slope. The experiment was conducted at $25 \pm 3$ °C, which was close to ambient temperature.

The response curve was set to hyperbolic function $A = a\,[(A\mathrm{max} \times \mathrm{PPFD})/(b + \mathrm{PPFD})]$, where $A$max is the maximum $CO_2$ assimilation and a and b are the hyperbolic equation's parameters. Dark respiration (equation parameter a) and light point compensation (equivalent to the PPFD value where $A$ is zero) were computed using this function. A straight line (y = 1) was fitted to the highest points of the curve to establish the light saturation point.

### 2.5. Productive and Structural Traits

After 120 days, plants were collected in February and March 2021 and transported to the Laboratory of Medicinal Plants (FCA UNESP) for the following tests: first pod insertion height (cm): measured with the assistance of a measuring tape from ground level to the first pod on a sample of 10 plants taken from the usable area; plant height (cm): measured with a measuring tape from ground level to the end of the tallest leaf from the same sample; number of pods per plant: counting the number of pods per plant from the same sample; and immature grain production per plant (g plant$^{-1}$): calculated using the number of pods per plant, the average number of seeds per pod, and the average weight of 100 edamames.

### 2.6. Statistical Analysis

Data were previously submitted to Bartlett's homogeneity and normality tests using the Shapiro–Wilk test. The analysis of variance (F-test) was done using R Studio software [27] after testing for normality and homogeneity, and means were compared using

the Tukey test with a 5% probability. The SAS 9 statistical program was used to adjust the hyperbolic function of the light response curves. MetaboAnalyst 4.0 software was employed for the multivariate analysis.

## 3. Results

### 3.1. Daily Gas Exchanges

In terms of the gas exchange parameters measured during the day, the vegetable and type soybean cultivar grain cultivars behaved differently (Figure 1). A divergent trend was observed between the vegetable soybean cultivar BRS 267 and type soybean cultivar grain 58HO124 EP RR in regards to net assimilation dynamic (12:00 p.m.), stomatal conductance (10:00 a.m.), intercellular $CO_2$ concentration (6:00 p.m.), and transpiration (10:00 a.m., 2:00 p.m.). These can be attributed to the specificity of the metabolic pathways (Table 2).

With regards to $gs$ (Figure 1B), the cultivars behaved differently (until 12:00 p.m.); after that, there was a decrease in this variable, which was influenced by the increase in temperature and decrease in relative humidity of the air (until 4:00 p.m.) (Table 2); however, this behavior was observed (after 4:00 p.m.) due to the lower PFFD (Table 2). The type soybean cultivar grain 58HO124 EP RR had the lowest $Gs$ values, which might indicate a tactic to prevent water loss by partly shutting the stomata. The $Gs$ peak for the vegetable type BRS 267 was about 0.53 mol m$^{-2}$ s$^{-1}$ around 10:00 a.m., when relative humidity and temperature were at their maximum (Table 2).

$Ci$ was always greater in vegetable-type BRSMG 790A, except at 10:00 a.m. (Figure 1C). Type soybean cultivar grain 58HO124 EP RR showed the lowest $Ci$ values during the day (except at 6:00 p.m.), which might imply that $CO_2$ obtained by stomatal opening was utilized for carboxylation. The peak of $Ci$ for the vegetable-type BRS 267 was at 10:00 a.m., the same time as the maximum $Gs$. The cultivars vegetable-type BRS 267 and type soybean cultivar grain 58HO124 EP RR had higher transpiration values throughout the day, with peaks at various times (Figure 1D). Even though the $Gs$ were greater at times, the cultivar vegetable-type BRSMG 790A demonstrated less water loss than the others.

The peak of the $E$ results varied across the cultivars studied, with greater temperature, lower relative humidity, and higher PFFD being linked to increased transpiration rate, suggesting that the cultivars studied exhibited distinct physiological responses to climatic conditions. The type soybean cultivar grain 58HO124 EP RR and vegetable-type BRS 267 achieved the maximum $E$, roughly 8.0 mol $H_2O$ m$^{-2}$ s$^{-1}$ at 12:00 p.m. and 2:00 p.m., respectively.

Assuming that the data were mostly within the same range, the water use efficiency was comparable between the vegetable and type soybean cultivar grain (Figure 2A). When compared to vegetable-type BRSMG 790A and type soybean cultivar grain 58HO124 EP RR, vegetable-type BRS 267 performed better at 6:00 p.m. since vegetable-type BRS 267 behavior was connected to reduced transpiration (E) and greater $CO_2$ assimilation (A) around 6:00 p.m., with a water use efficiency of roughly 7.1 µmol $CO_2$ (mol $H_2O$)$^{-1}$.

Plants of cultivar vegetable-type BRSMG 790A were less efficient than those of vegetable-type BRS 267 and type soybean cultivar grain 58HO124 EP RR in terms of $A/Ci$ (Figure 2B). These results are consistent with those for $CO_2$ assimilation rate, which occurred at 12:00 p.m. for type soybean cultivar grain 58HO124 EP RR and 2:00 a.m. for vegetable-type BRS 267.

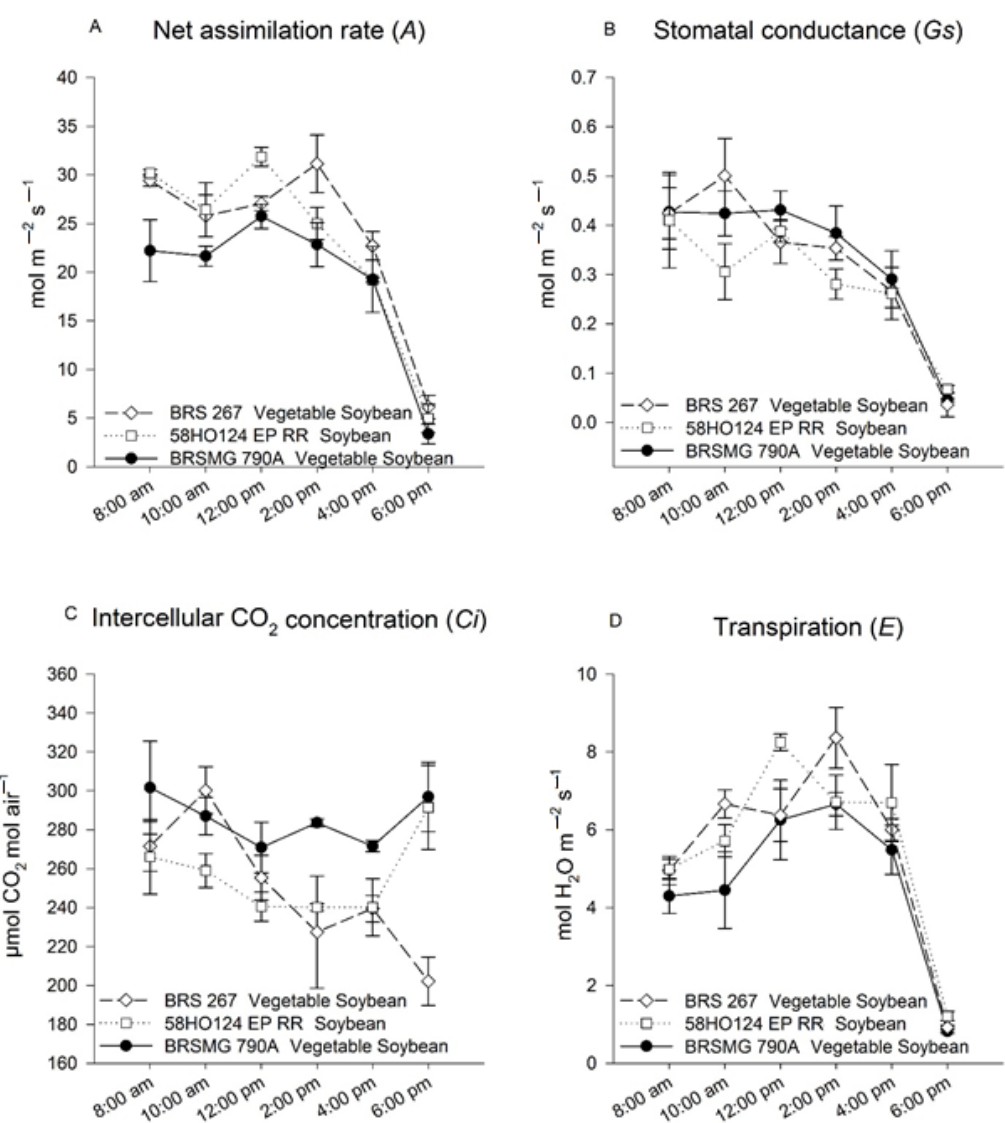

**Figure 1.** Net $CO_2$ assimilation rate ($A$, $\mu$mol m$^{-2}$ s$^{-1}$) (**A**), stomatal conductance ($Gs$, mol m$^{-2}$ s$^{-1}$) (**B**), intercellular $CO_2$ concentration ($Ci$, $\mu$mol $CO_2$ mol air$^{-1}$) (**C**), and transpiration ($E$, mol $H_2O$ m$^{-2}$ s$^{-1}$) (**D**) of vegetable and conventional soybean cultivars. Botucatu, SP, Brazil, 2021.

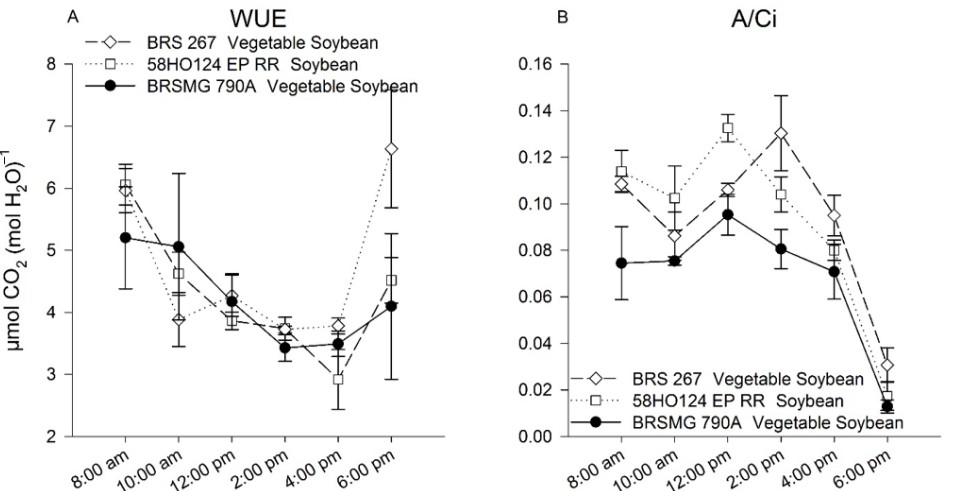

**Figure 2.** Water use efficiency (WUE, $\mu$mol $CO_2$ (mol $H_2O$)$^{-1}$) (**A**) and carboxylation efficiency ($A/Ci$) (**B**) of vegetable and type soybean cultivar grain soybean cultivars. Botucatu, SP, Brazil, 2021.

### 3.2. CO₂ Assimilation Response Curve as a Function of PPFD

PFFD had a varied effect on different cultivars (Figure 3). Table 3 shows significant differences in light compensation point, light saturation point, and dark respiration amongst cultivars, indicating that both vegetable and type soybean cultivar grain required distinct PPFD to absorb $CO_2$ and accumulate dry matter.

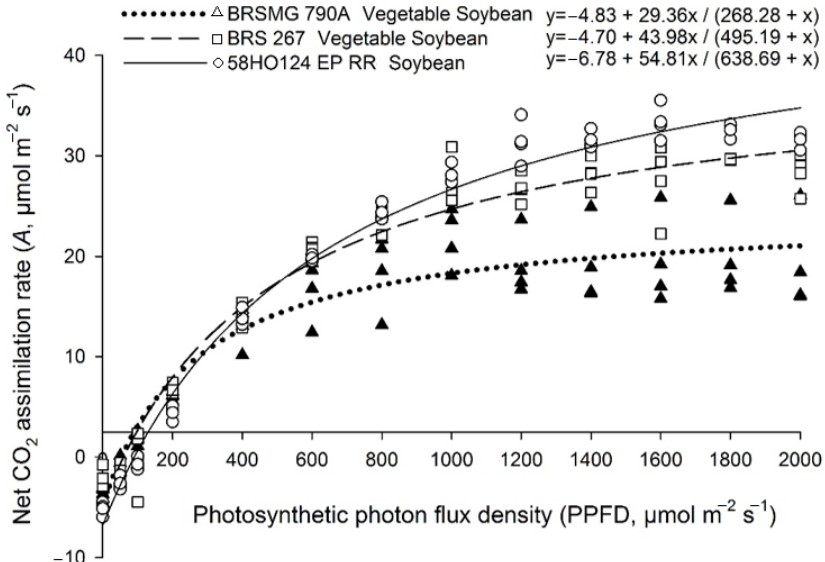

**Figure 3.** Net $CO_2$ assimilation rate ($A$, µmol m$^{-2}$ s$^{-1}$) as a function of photosynthetic photon flux density (PPFD, µmol m$^{-2}$ s$^{-1}$) of vegetable and type soybean cultivar grain. Botucatu, SP, Brazil, 2021.

**Table 3.** Light compensation point (µmol m$^{-2}$ s$^{-1}$), light saturation point (µmol m$^{-2}$ s$^{-1}$), and dark respiration (µmol m$^{-2}$ s$^{-1}$) of vegetable and type soybean cultivar grain soybean cultivars. Botucatu, SP, Brazil, 2021.

| | Light Saturation Point | Light Compensation Point | Dark Respiration |
|---|---|---|---|
| Cultivars | | µmol m$^{-2}$ s$^{-1}$ | |
| BRSMG 790A | 1619 c | 36.1 c | −4.5 b |
| BRS 267 | 2125 b | 46.4 b | −4.6 b |
| 58HO124 EP RR | 2430 a | 69.3 a | −6.5 a |
| F | 75.3 * | 41.7 * | 85.2 * |
| CV (%) | 11.4 | 8.6 | 4.1 |

Tukey's test with a 5% probability shows that the means following the same letter do not differ substantially. * Significant at a 5% probability ($p > 0.05$).

When compared to the others, vegetable-type BRSMG 790A presented a propensity to photosynthetic saturation, with a PFFD of about 1.619 µmol m$^{-2}$ s$^{-1}$, while vegetable-type BRS 267, another vegetable type, had a greater saturation of 2.125 µmol m$^{-2}$ s$^{-1}$, indicating that it may be more resistant to solar radiation stress, even within the same category. Furthermore, the highest light saturation point was observed in type soybean cultivar grain 58HO124 EP RR, with 2.430 µmol m$^{-2}$ s$^{-1}$.

When compared to vegetable-type BRSMG 790A, the light compensation point followed that of saturation with type soybean cultivar grain 58HO124 EP RR and vegetable-type BRS 267, as both required higher PFFD to begin $CO_2$ assimilation. With regards to type soybean cultivar grain 58HO124 EP RR, dark respiration was also superior to the vegetable types as they did not vary from one another in terms of dark respiration.

### 3.3. Productive and Structural Characteristics

All variables of productive performance tested, such as the first pod insertion height, plant height, number of pods per plant, and production of immature grains per plant, showed significant variations between treatments (Table 4).

**Table 4.** Averages from the first pod insertion height (FPIH), plant height (PH), number of pods per plant (NPP), production of immature grains per plant (PIGP). Botucatu, SP, Brazil, 2021.

| Cultivars | FPIH | PH | NPP | PIGP | Mean and SD ** |
| --- | --- | --- | --- | --- | --- |
| | (cm) | (cm) | - | (g pl$^{-1}$) | |
| BRS 267 | 14.6 a | 138.7 a | 139.0 b | 270.2 b | 249.5 ± 81.1 |
| BRSMG 790A | 17.0 a | 123.0 b | 144.1 b | 547.6 a | 603.4 ± 388.7 |
| 58HO124 EP RR | 6.6 b | 79.3 c | 233.8 a | 339.1 b | 351.0 ± 174.9 |
| F | 43.7 * | 182.3 * | 5.6 * | 9.2 * | - |
| CV (5%) | 10.3 | 6.3 | 36.8 | 34.8 | - |

\* Significant at the 0.05 probability levels. Means in the same column do not differ according to the Tukey test at the 5% level of probability. ** Mean and SD of production of immature grains per plant.

For the first pod insertion height (FPIH), the cultivars vegetable-type BRS 267 (14.6 cm) and vegetable-type BRSMG 790A (17 cm) had the highest averages, which did not vary statistically. In terms of plant height (PH), the highest average was 138.7 cm for vegetable-type BRS 267 (i.e., a sort of soybean that is ready to eat as a vegetable), which was 15.65 cm higher than the plants of vegetable-type BRSMG 790A (144.1 cm) and 59.4 cm higher than the type soybean cultivar grain 58HO124 EP RR (79.3 cm).

Moreover, the type soybean cultivar grain 58HO124 EP RR exhibited a greater average number of pods per plant (NPP) (233.8). In terms of production of immature grains per plant (PIGP), vegetable-type (BRSMG 790A) had the highest average (547.6 g pl$^{-1}$).

### 3.4. Multivariate Analysis

The graphical data depicts the agronomic performance parameters and their relevance to the 14-h gas exchange (Figures 4 and 5). The PCA projection obtained for the three genotypes studied, based on the 9 parameters, explains 56.4% of the variance (Figure 4). The components of vegetable-type BRS 267 and vegetable-type BRSMG 790A were comparable; however, there was minimal resemblance with the components of type soybean cultivar grain 58HO124 EP RR (Figure 4A).

There was a direct connection between $gs$ and $Ci$ in Biplot PC1 × PC2 (Figure 4B). These, in turn, were inversely proportional to the PIGP and the other variables. Moreover, WUE was also unrelated to PH and FPIH. However, a resemblance between NPP and all the variables can be seen on the biplot, indicating that physiological parameters were related to productive performance characteristics and vice versa.

Cluster analysis is the segmentation of a heterogeneous population into a set of more homogenous subgroups (Figure 5A). There are no pre-defined classifications in the grouping; the components are classified according to similarity.

The Pearson graph may be used to find correlations that are both directly and inversely proportional (Figure 5B). The variables PIGP and $E$ were shown to be directly proportional and somewhat linked, and the same can be seen in $gs$-related $Ci$. Furthermore, $Ci$ was shown to have inversely proportional and strong correlations with $E$. Aside from that, $E$ had an inversely proportional and moderately associated relationship with $gs$, while PIGP had an inversely proportional and moderately correlated relationship with $Ci$.

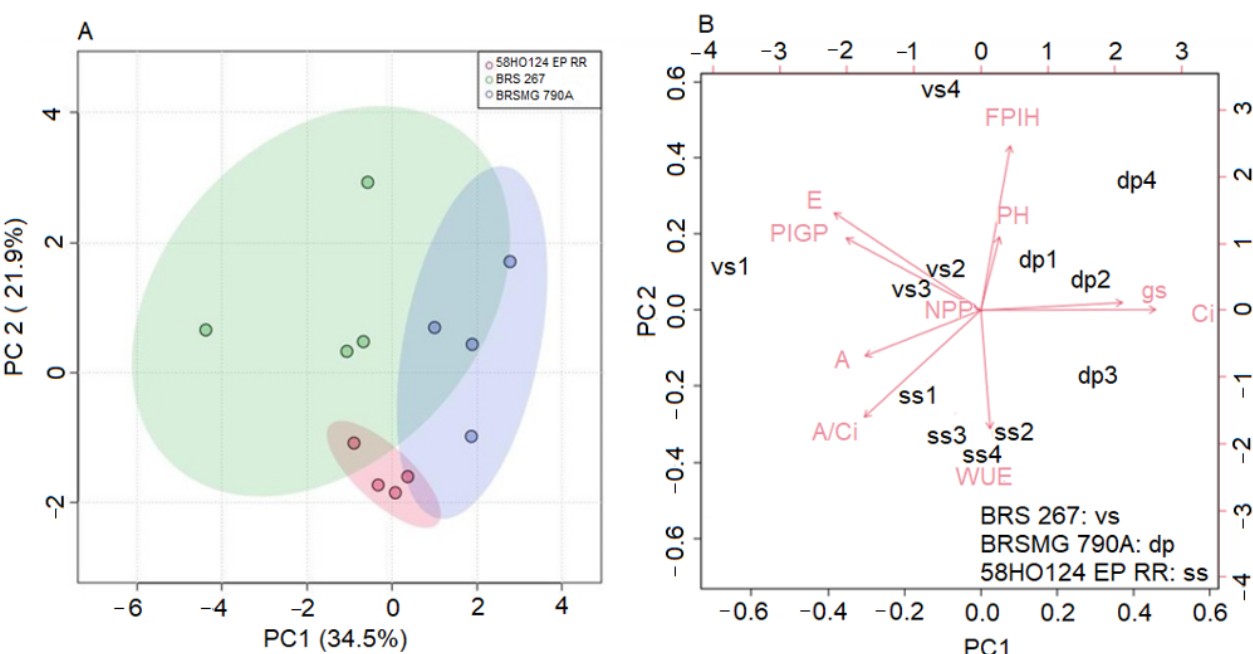

**Figure 4.** Score plot (**A**) and biplot (**B**) of the first two principal components of agronomic traits and gas exchange of soybean cultivars. Botucatu, SP, Brazil, 2021.

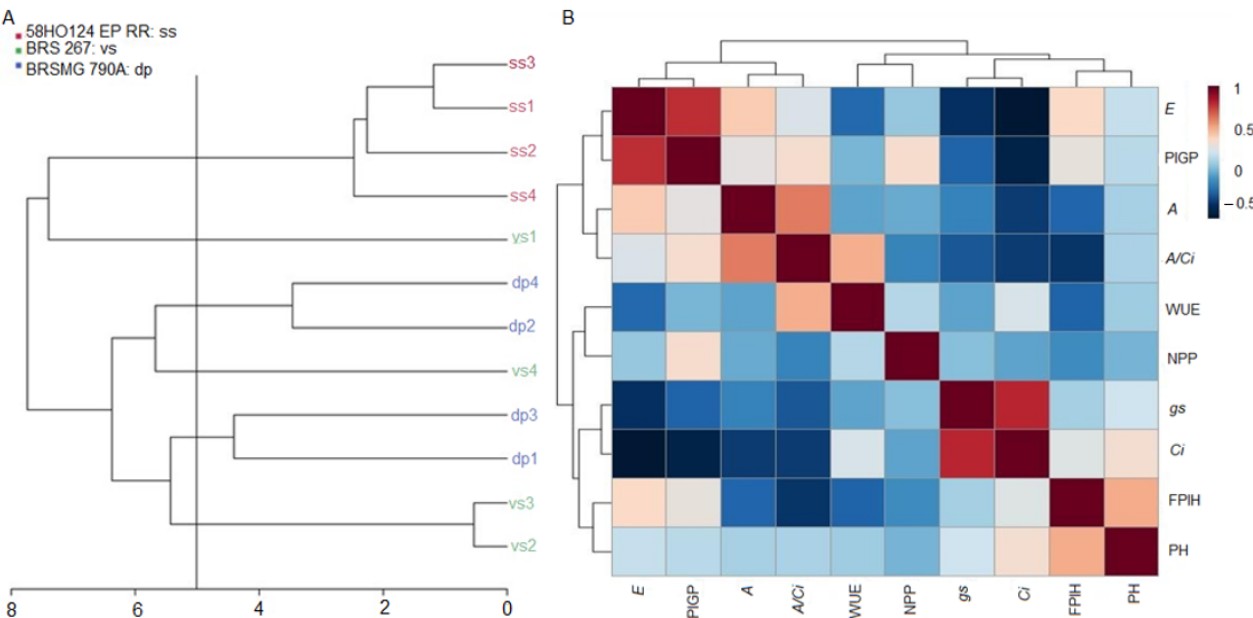

**Figure 5.** Cluster analysis (**A**) and Pearson graph (**B**) of agronomic characteristics and gas exchange of soybean cultivars. Botucatu, SP, Brazil, 2021.

## 4. Discussion

Analyses of the net assimilation rate and the growth rate of the soybean crop revealed that the crop undergoes two peaks of photosynthetic activity, one during flowering (R1 and R2 stages) and another during pod fill (R 5.1 and R 5.2 stages) [28]. For this, as well as the early harvest of vegetable soybeans, we chose to investigate gas exchange and light responses in the R1–R2 stage.

Photosynthesis is a complex physiological and biochemical process that is hampered by a variety of environmental variables, including PAR, temperature, humidity, and $CO_2$ concentration [29]. The physiological behavior of vegetable and type soybean cultivar grain

cultivars was demonstrated in this study through the changes in the PFFD, temperature, and relative humidity throughout the day.

Based on the light response curve, the cultivars revealed distinct timings for the peak of *A* (Figure 1A). This is mostly owing to the PFFD (Figure 3). The peak of $CO_2$ assimilation occurred in the vegetable-type BRS 267 at 2:00 p.m., when the PFFD was 1570 µmol m$^{-2}$ s$^{-1}$. The cultivar vegetable-type BRSMG 790A and type soybean cultivar grain 58HO124 EP RR had the highest PFFD at 12:00 p.m., that is, 2150 µmol m$^{-2}$ s$^{-1}$. Reduced shadowing and increased PAR transmittance to 1636.8 µmol.m$^{-2}$.s$^{-1}$ improved the photosynthetic properties of soybean plants, according to research measuring light intensity in intercropping soybean and corn crops [30]. The soybean crop has been shown to be PAR-demanding, and the selection of conventional- or vegetable-type cultivars for specific locations or farming forms (consortium with large crops or vegetables) should consider the light intensity of the growing environment.

When considering the production of vegetable soybean in different areas, it is feasible to identify vegetable-type BRS 267 in regions with lower solar radiation and vegetable-type BRSMG 790A in places with higher light intensity. Because they have comparable physiological behavior in terms of gas exchange and light response curve, the cultivar vegetable-type BRSMG 790A is suited for locations previously designated for the growth of the type soybean cultivar grain 58HO124 EP RR.

The light compensation point denotes that the photosynthetic process assimilated the same quantity of $CO_2$ emitted by respiration, i.e., *A* was equal to respiration at this point. The highest light compensation point was found in the type soybean cultivar grain 58HO124 EP RR, indicating that this cultivar had greater respiration than the vegetable types. Another feature of the light curve where the type soybean cultivar grain 58HO124 EP RR fared better than the others is the light saturation point. Besides having the best respiration rate of all, this cultivar also proved to be more resistant to direct sunlight than the others. Furthermore, vegetable-type BRSMG 790A tended to saturate photosynthesis at lower PFFD values than vegetable-type BRS 267 when comparing both vegetable types. This suggests that the vegetable-type BRS 267 cultivar plants made greater use of incoming solar energy, were more resistant to photo inhibitory stress, and took longer to restrict $CO_2$ uptake. The vegetable-type BRSMG 790A, on the other hand, was more susceptible to environmental factors. There is no substantial increase in photosynthesis when exposed to strong sunlight. In this case, photosynthesis has been saturated by radiation, and photochemical processes are no longer limiting the $CO_2$ assimilation rate [31]. When compared to cultivars vegetable-type BRS 267 and type soybean cultivar grain 58HO124 EP RR, the electron transport rate and RuBisCO enzyme activity became more limiting in cultivar vegetable-type BRSMG 790A.

From the results of gas exchange, it was possible to show the relationship between *gs*/*E* and environmental stress; the cultivars exhibited different behaviors on environmental stress tolerance. With increasing temperature and decreasing relative humidity throughout the day, cultivars vegetable-type BRS 267 and type soybean cultivar grain 58HO124 EP RR displayed lower *gs* and greater *E*, indicating sensitivity to weather conditions, but vegetable-type BRSMG 790A maintained constant *gs* and lower *E* than the others. In non-propagated cucumbers, a similar relationship among environment, *gs*, and *E* was seen; they appear to be more susceptible to environmental factors since they exhibited reduced stomatal conductance because of a rise in temperature and a decrease in relative humidity [32].

Water availability is one of the elements that might affect crop output variations across years or regions of farming. By understanding it, more water-efficient cultivars should be sought, and the cultivars in this study exhibited comparable performance when it came to the correlation between $CO_2$ absorbed and $H_2O$ released into the atmosphere.

With regards to first pod insertion height, [33] achieved greater averages of 16 and 12 cm for the first year of edamame cultivation and an average of 13 cm for the second year of cultivation, with these results being similar to those reported in this current study.

Although distinct soybean-vegetable cultivars were employed, the resemblance is due to the right selection of cultivars based on the edaphoclimatic conditions of each production location in Shorter, AL, USA, and Sao Paulo, Brazil, allowing each material to exhibit its full inherent genetic potential.

In terms of plant height, the findings differ from those of [34], who obtained the greatest average plant height of 88.3 cm when analyzing optimal plant density for edamame production. The adoption of automated edamame harvesting accounts for this discrepancy. The plant design is a decisive element in this type of harvest, and it may either help or hinder the effectiveness of this process [34].

As a result, cultivars with shorter initial pod insertion height and plant height can be appealing for commercial production with automated harvesting. Thus, mechanical harvesting systems for vegetable soybean production are uncommon in Brazil since this technology is not suitable for harvesting grains with high moisture content yet [16]; therefore, the crop is harvested entirely by hand, cutting the still green plant [35].

However, cultivars with a higher first pod insertion height and plant height provide better ergonomics for farmers and rural laborers, making harvesting easier and increasing the amount of area harvested per day. In this way, the vegetable-types BRS 267 and BRSMG 790A stood out in this setting.

Nevertheless, low-growing genotypes have fewer pods because the number of pods in vegetable-type cultivars is proportional to the crop size of the crop [36]. However, the shorter cultivar, the type soybean cultivar grain 58HO124 EP RR, had the greatest average for this feature. This contrast is owing to the cultivar's intended use as a grain-type soybean with traits that make mechanical harvesting easier, such as size, lower height of first pod insertion, and production characteristics, such as a high number of pods, bearing in mind that plant heights more than 80 cm and pod insertion heights less than 10 cm in conventional soybean cultivars indicate losses during mechanical harvesting [37].

Santos et al. (2013) [36] observed 17.90 (in 2010) and 24.80 (in 2011) pods per plant for vegetable-type BRS 267, which differs from the current study. Such difference is due to the plants' superior adaptability to the warm humid temperate temperature (Sao Paulo, Brazil) when compared to the hot and humid tropical climate (Bahia, Brazil). Another study [38] assessed 19 vegetable soybean genotypes in Jay and Citra, Florida, and found higher average pod counts of 135.9 in Jay and 109.7 in Citra, which are comparable to the cultivars in this study.

Increases in growth potential, dry matter accumulation, yields, and final product quality can all be attributed to improved $CO_2$ assimilation and carboxylation efficiency. The outcome of photoassimilates, on the other hand, is determined by the genetic potential of each crop and its interaction with the environment. As a consequence, vegetable-type BRSMG 790A produced more immature grains ($g \cdot plant^{-1}$) than the other cultivars tested, probably due to its dual aptitude; while having lower $A$ and $A/Ci$ values than the other ones, this cultivar was able to convert absorbed $CO_2$ into grain mass.

The type soybean cultivar grain 58HO124 EP RR had the greatest $A$ and $A/Ci$ values at certain times of the day. Due to the material's own genetic potential and aptitude, this cultivar directs the photoassimilates to floral emission, which increases the number of pods but does not direct much to promote plant development. However, even with more pods, grain filling remains the same, and this cultivar does not generate immature grains ($g \cdot plant^{-1}$).

On the other hand, the vegetable-type BRS 267, which had higher $A$ and $A/Ci$ values than the type soybean cultivar grain 58HO124 EP RR, directed the $CO_2$ fixed for growth, resulting in a higher height than the others. This cultivar did not convert into grain filling even with the highest vegetative vigor.

The influence of gas exchange on the morphological and productive features of the studied soybean cultivars was demonstrated through multivariate data analysis (Figure 4). For example, $A/Ci$ is proportional to PGIP (Figure 4D), and carboxylation efficiency evaluates how fast and effectively a plant assimilates $CO_2$, which may be employed for future

production gains. This significant correlation between $A/Ci$ and soybean yield was also reported by [39]. According to the clustering analysis, the vegetable-types BRS 267 and BRSMG 790A inferred greater similarity because both have a vegetable purpose and express traits indicating this aptitude, and they have less similarity with the type soybean cultivar grain 58HO124 EP RR, which was not grouped due to its one-of-a-kind appropriateness as a grain soybean purpose.

The establishment of low-impact organic production practices management for vegetable-type soybean with low cost and high yield would be one of the major obstacles for research, according to the review authors, since this crop is yet undeveloped and lacking in data [40]. Because of this, encouraging responses about the organic growth of this crop, with promising outcomes, have been obtained, which will undoubtedly serve to motivate and direct future work on edamame production.

## 5. Conclusions

The soybean grain and vegetable types of soybean showed different behaviors in physiology and yield. The type soybean cultivar grain 58HO124 EP RR showed greater assimilation of $CO_2$; however, the photoassimilates were directed to floral emission, because such features are inherent in its ability.

The vegetable-type BRS 267, on the other hand, did not exhibit promising production outcomes, directing its photoassimilates to vegetative growth, in addition to having an excellent pod insertion height, which improves manual harvesting efficiency.

Finally, vegetable-type BRSMG 790A produced the most immature grains per plant while also having the greatest first pod insertion height, being the best in converting photoassimilates for edamame production.

**Author Contributions:** Conceptualization, investigation, methodology, project administration, writing—original draft, L.M.R.; investigation, methodology, data curation, formal analysis, writing—original draft, E.S.A.; investigation, methodology, data curation, C.S.N., G.d.C.F. and R.I.R.; supervision, visualization, writing—review and editing, F.P.G.B.; validation, visualization, writing—review and editing, J.D.R. and E.O.O. All authors have read and agreed to the published version of the manuscript.

**Funding:** This research received no external funding.

**Institutional Review Board Statement:** Not applicable.

**Informed Consent Statement:** Not applicable.

**Acknowledgments:** The first author acknowledgments the Conselho Nacional de Desenvolvimento Científico e Tecnológico (CNPq) for awarding him a master's scholarship.

**Conflicts of Interest:** The authors declare no conflict of interest.

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
