# Peer review of "Assessment of the Physiological Response and Productive Performance of Vegetable vs. Conventional Soybean Cultivars for Edamame Production"

_agronomy, doi:10.3390/agronomy12061478_

Round 1

Reviewer 1 Report

The authors studied the physiological responses of three soybean cultivars for edamame. 

Below the authors can find suggestions that aim to improve the manuscript quality.

Line 74: "vagetarianism" or vegetarianism? 

Line 84: "aim to compare"

Line 92: FCA means Farm Lageado?

Lines 92-94: Suggestion: "Universidade Estadual Paulista (FCA/UNESP), Botucatu campus, in the state of São Paulo (22º51'S; 48º26'W, 786m a.s.l.)." Instead of detailing the coordinates and altitude.

Line 97: Use the degree symbol (ALT+167)

Line 134: About the Law 10831 (Brazilian Organic Law). 

s far as I know, it is not allowed the use of any transgenic products in the organic farming in Brazil according to the Organic Law nº10831 and Technical Regulations (NI 46 and NI 17). In addition, in other countries the use of transgenic cultivars are very restrictive if not forbiden according to their Organic Law.

I raised that question because the authors used 58HO124 EP RR cultivar in the experiments, which is a transgenic cultivar. 

Then, I suggest that the authors could use the term "low impact agricultural practices" or similar, instead of "organic", because the readers could be confused if it is allowed to grow an "organic-transgenic edamame".

Line 157: In my opinion, the authors could change the use of the term "conventional cultivar" in the text because it may induce the readers to think that "58HO124 EP RR" is not a genetically modified soybean cv. Then, this is a cultivar commonly used for grain and animal feed. Not a non-transgenic cultivar. 

Figure 4. Authors must use a greater font size in the plots, PCA and dendrogram. It is not possible to read the content.

Line 458: Here it is better to cite the author's last name to avoid iniciating a paragraph with the reference number

Attached is the version of the manuscript with the suggestions

Author Response

Dear reviewer!

Line 74: corrected

Line 84: corrected

Line 92: FCA stands for Faculty of Agricultural Sciences. Fazenda Lageado is the name of the place where FCA is based.

Lines 92-94: corrected

Line 97: corrected

Line 134: Thanks for the correction, changes have been made.

Line 157: Changed expression.

Figure 4. Graphs redone with larger letters and numbers.

Line 458: Change made.

Reviewer 2 Report

The manuscript with the title “Assessment of the physiological response and productive performance of vegetable vs. conventional soybean cultivars under organic management for edamame production” provides a comparative assessment of some soybean cultivars for edamame production under organic management. Edamame, more widely eaten in Asia, represents an emerging food option for healthy lifestyle in Brazil with good prospects for crop expansion in the future. Results are interesting, novel and have agronomic relevance. The authors evaluated the physiological response and key agronomic traits because these are related to the productivity performance and therefore these are relevant for the feasibility of the crop.  

All my concerns expressed in the previous report have been addressed. The manuscript has increased in readability.  Only some English style and syntax issues remain to be addressed. 

Best regards.

Author Response

Dear reviewer! Thanks for the feedback, we made the requested changes.

Round 2

Reviewer 1 Report

Dear authors, the use of the term "organic" in the manuscript must be revised.

According to the Law nº10,831/2003; regulated by Normative Instruction nº46/2011; NI nº17/2014 and ordinance number 52/2021, it is strictly forbidden the use of genetically modified organism (GMO) in Brazil. 

Where the authors used the term "organic" for describe the agricultural management or agricultural system which the plants were grown this should be revised because it is not possible to grow organics with GMO seeds (according to the Law). 

The authors must keep in mind that the organic creditation/certification is not for the plants but the area that these plants are cultivated. Then, if  forbidden inputs (synthetic herbicides, insecticides, fertilizers, seeds) were applied on the area, it is not appropriated for certification, according to the law.

Below there are the main points. Other points you can find in the revised version of the MS. In the attached file I delete several parts that the use of the term organic could be misused. In these parts, instead of deleting, the authors could substitute to "cultivated with fertilizer and phytosanitary products allowed for the organic agriculture" or a similar sentence.

For the title I suggest that could be reformulated as follow:

"Assessment of the physiological response and productive performance of vegetable vs. conventional soybean cultivars for edamame production"

In Material and Methods section:

Lines 136-139: "Cultural treatments for low impact agricultural practices organic production management (such as manual weeding, disease, and pest control) were implemented in line with Law No. 10,831 [24] and the Technical Regulation of Normative Instruction 46 [25], which were supplemented by IN 17 [26]".

In this sentence must be added the explanation on why the authors used a GMO cultivar. An alternative could be add a sentence that "plants were grown in line with Law..., except for the use of a GMO cultivar". To make it clear to the readers that the Law above mentioned was followed properly, except for the use of the GMO (which is forbidden).

Other option (my first recommendation) is to use the sentence (or adapted to the context) "cultivated with fertilizer and phytosanitary products allowed for the organic agriculture" when the authors refer to the agricultural system used in the experiments.

Author Response

May , 2022

Dear Editors-in-Chief:

thank you for receives our work No. agronomy-1715843, entitled “Assessment of the physiological response and productive performance of vegetable vs. conventional soybean cultivars for edamame production” for consideration for publication in the Agronomy MDPI. The manuscript was carefully revised, and all the reviewers’ comments were attended as follow:

Reviewer #1:

According to the Law nº10,831/2003; regulated by Normative Instruction nº46/2011; NI nº17/2014 and ordinance number 52/2021, it is strictly forbidden the use of genetically modified organism (GMO) in Brazil: correct.

Where the authors used the term "organic" for describe the agricultural management or agricultural system which the plants were grown this should be revised because it is not possible to grow organics with GMO seeds (according to the Law): the term “organic” was removed.

For the title I suggest that could be reformulated as follow:

"Assessment of the physiological response and productive performance of vegetable vs. conventional soybean cultivars for edamame production": reformulated.

The authors must keep in mind that the organic creditation/certification is not for the plants but the area that these plants are cultivated. Then, if  forbidden inputs (synthetic herbicides, insecticides, fertilizers, seeds) were applied on the area, it is not appropriated for certification, according to the law: the term “organic” was removed.

Below there are the main points. Other points you can find in the revised version of the MS. In the attached file I delete several parts that the use of the term organic could be misused. In these parts, instead of deleting, the authors could substitute to "cultivated with fertilizer and phytosanitary products allowed for the organic agriculture" or a similar sentence: dear reviewer, no fertilizers or phytosanitary products were used in the experimental area. The mistake in using the term organic was because we used a GMO cultivar. In this case, the changes were made.

Lines 136-139: rewritten sentence, as suggested.

Lines 61-62: “The low impact agricultural practices is an example of natural management that is used by most vegetable growers…”.

Line 104: “for edamame production”.

Lines 505-506: “low impact practices”.

Thank you for your consideration!

Sincerely,

Laura Matos Ribera, Me.

This manuscript is a resubmission of an earlier submission. The following is a list of the peer review reports and author responses from that submission.

Round 1

Reviewer 1 Report

The manuscript with the title “Physiological and productive performance of vegetable and conventional soybean cultivars under organic management” provides a comparative assessment of some soybean cultivars for edamame production under organic management. Results are interesting and have agronomic relevance. The authors evaluated the physiological response and key agronomic traits because these are related to the productivity performance and therefore these are relevant for the feasibility of the crop. Below I gave some comments and suggestions for authors, in the hope that these will help them to improve their manuscript. 

Title: can be improved to reflect better the content. I propose something like: “Assessment of the physiological response and productive performance of vegetable vs. conventional soybean cultivars under organic management for edamame production”

Abstract

Start the abstract with a phrase that immediately can highlight the importance of the study such as “There is an increasing interest for organic cultivation but feasibility depends on the precise knowledge on the productivity of different genotypes under such management. Because there is a close relationship between plant physiological response and crop performance, the current study aimed to evaluate the photosynthetic efficiency and productive performance of vegetable versus conventional soybean cultivars grown under organic conditions. …” Please mention in the abstract the general trend and pattern observed, that might be relevant for other researchers as well as for the international audience as well. 

Introduction

The Introduction section needs to be expanded. This section starts with the importance of this crop, and this is OK. Please also find sources to cite the average production obtained (kg/ha) and some statistics on the topic (something more about this crop). How important is this crop in Brazil? Or does it present export potential as well? Please address current challenges for organic cultivation of this crop or in general.  Please give some brief specifications regarding legislation/guidelines on organic cultivation in Brazil or otherwise. What are the prospects for this crop and the current national/international trends? At the end of the Introduction section, please clearly express the aim and objectives of the research.  

Material and Method 

Some paragraphs from “Treatments” section should be merged with “Growing conditions” section because they refer to the study site, and cultivation. I refer here to the Lines 106-115. 

Replace “treatments”, section title with “Experimental design”, and present herein the factors and experiment organization. Please mention the rationale of choosing these genotypes for study and provide brief description of these cultivars – particularly how widely cultivated are they if such information exists. Are these autochthonous cultivars adapted to local conditions or recommended/designated as suitable for organic cultivation systems?  These are important for the agronomic relevance of your study. 

Results

The way results are presented could be improved. Please also present the general pattern observed and highlight the important aspects. For example, “It was observed a divergent trend between the vegetable soybean cultivar (BRS 267) and conventional soybean (58HO124) in regards with net assimilation dynamic (12:00 pm), stomatal conductance (10:00 am), intercellular CO2 concentration (6:00 pm) and transpiration (10:00 am, 2:00 pm). These can be attributed to specificity of the metabolic pathways etc…” 

Table 4 - please present the productivity results (agronomic traits) as mean ± SD or SE. Mentioning the Range (X max - X min) could also be very suggestive for these traits in the text. 

Line 271, I suggest to express the results such as “PCA projection (Figure 4a, 4b) obtained for the three genotypes studied based on the 9 parameters, explains 56.4% of the variance.” 

Figure 4 – the writing is too small, use larger font. You can present 4c and 4d as separate, independent figures. Make them larger or use larger font. 

Discussion

Although the information presented overall is good, the discussion section needs to be made more cohesive. Please do not use small paragraphs and do not jump from one idea to another. The entire discussion section should have a logical organization from start to finish to make it easy for the reader to follow-through the ideas presented. I suggest the following structure: 1) General remarks about the study, 2) comparison with results obtained by other authors in order to have a base of reference for the values presented here (are these small or high values? in reference with other studies), 3) agronomic significance of the results and subsequent implications, 4) challenges, recommendations and future prospects on this topic. 

Conclusions

Conclusions should mirror the objectives. If for reaching your aim to had 2 objectives defined, I assume (Obj1 = comparative assessment of physiological response and Obj2 = agronomic traits) then in conclusions you should answer exactly to these objectives. Please formulate clearly the aim and objectives and then write the conclusions to answer the objectives proposed. Do not forget to include some recommendations and general conclusion referring to the trend observed that might be helpful for other researchers abroad and international agronomists. 

References

Reference list should include more recent studies. There have been published studies on the topics related to this, within last 1-2 years (just give a google scholar search) you will find a handful of suitable papers that you could cite in introduction and discussion to reach at least about 40 references in your list. 

Best Regards.

Reviewer 2 Report

The authors studied photosyntesis and production variables of soybean under organic management. The manuscript has merit. Organic soybean finds diverse difficulties through crop production cycle. There are few examples of successfully large-scale organic soybean production areas.

I have some questions and suggestions to improve the manuscript quality.

Introduction

Line 35: What does moderate flavour mean? 

Line 42: In terms of nutritional properties are there differences between grain and soybean as a vegetable purposes?

Materials and methods

There are PT-BR terms that should be explained. For example: SB, CTC, V% and PRNT. 

For the insects incidence, I recommend that only scientific name could be cited. Some of common names used in the MS were literally translated to English. I am not sure if Aracanthus mourei, Diabrotica speciosa, and Magacelis sp. have an international stablished common name. I would only cite the scientific name instead. The authors can use the common name if desire. However these must be corrected. 

Suggestions: Anticarsia gemmatalis --> velvetbean caterpillar; Spodoptera frugiperda --> fall armyworm

Line 134: DFFF: Please, explain the meaning.

Results

Figure 4: What do E, A, PH, sh3, sh2.... stand for?
This should be explained. The algarisms in the figures are very small. I found difficulties for reading the numbers and letters.

I liked the discussion part. The results were carefully discussed.

Other minor points are cited in the commented version of the manuscript.
